# Electrical and Humidity-Sensing Properties of EuCl₂, Eu₂O₃ and EuCl₂/Eu₂O₃ Blend Films

## Pi-Guey Su * and Nok-Him Choy

Department of Chemistry, Chinese Culture University, Taipei 111, Taiwan; choynh516@gmail.com
* Correspondence: spg@ulive.pccu.edu.tw; Tel.: +886-2-2861-0511 (ext. 25332)

**Abstract:** Impedance-type humidity sensors based on EuCl₂, Eu₂O₃ and EuCl₂/Eu₂O₃ blend films were fabricated. The electrical properties of the pure EuCl₂ and Eu₂O₃ films and EuCl₂/Eu₂O₃ blend film that was blended with different amounts of EuCl₂ were investigated as functions of relative humidity. The influences of the EuCl₂ to the humidity-sensing properties (sensitivity and linearity) of the EuCl₂/Eu₂O₃ blend film were thus elucidated. The impedance-type humidity sensor that was made of a 7 wt% EuCl₂/Eu₂O₃ blend film exhibited the highest sensitivity, best linearity, a small hysteresis, a fast response time, a small temperature coefficient and long-term stability. The complex impedance plots were used to elucidate the role of ions in the humidity-sensing behavior of the EuCl₂/Eu₂O₃ blend film.

**Keywords:** humidity sensor; EuCl₂; Eu₂O₃; EuCl₂/Eu₂O₃ blend film; complex impedance plots

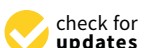

## 1. Introduction

Developing humidity sensors have attracted much interest because humidity is an important role in maintaining human health and an excellent quality of products [1–3]. Therefore, humidity sensors must have high sensitivity, a wide working humidity range, good linearity, fast response/recovery times, low hysteresis, good reversibility, stability and ease of fabrication for the mass production of humidity devices for using in food storage, industrial production and environmental monitoring [4,5]. Many materials, including ceramic, polyelectrolyte, organic polymer and composite materials, have been applied to humidity sensors [1,6–14]. Ceramic materials, including metal oxides, perovskite- and spinel-type oxides and their hybrid systems, have some superiority in function because of their good chemical stability, high heat resistance, good water resistance under high humidity, cost-effectiveness and fast response to the changes of humidity [7,15], which means they can be applied to humidity detection. The humidity-sensing properties of ceramic humidity sensors is strongly influenced by the surface activity and the porous structure of the ceramic materials [15]. Therefore, many reports focused on researching the microstructure and morphology of ceramic materials and doping various dopants to tune the physico-chemical properties of ceramic materials [6,16].

The rare earth elements (i.e., lanthanides) could be considered as active cocatalysts and dopants for the improvement of new substances with appealing gas-sensing applications because of their 7f orbitals awarding special electronic properties [17–25]. Zhong et al. [17] fabricated Eu₂O₃-doped In₂O₃ using the sol-gel method for detecting H₂S gas. Stănoiu et al. [18] fabricated ZnO–Eu₂O₃ binary oxide for sensing NO₂ gas under humid condition. Wang et al. [19] fabricated Eu-doped SnO₂ nanofibers for sensing acetone gas. Ortega et al. [20] fabricated Eu₂O₃-doped CeO₂ for sensing CO gas. Er et al. [21] fabricated rare earth metals (Y, Ru and Cs)-doped ZnO thin films for sensing NH₃ gas at room temperature. Jing et al. [22] fabricated a PANI/Eu³⁺ nanofiber for sensing NH₃ gas. Costello et al. [23] fabricated Eu³⁺ ion-doped ZrO₂ for sensing volatile organic compounds (VOCs). Mokoena et al. [24] fabricated Eu³⁺ ion-doped NiO for sensing toluene gas. Shen et al. [25] fabricated Ce-doped ZnO nanowires for sensing ethanol. Zhang et al. [26] fabricated a humidity

sensor that was made of Eu-doped ZnO using the sol-gel method. Most literatures tend to explore the effects of rare earth ions and oxides doping on the enhancing gas-sensing properties. Recently, Wang et al. [27] fabricated a fast response humidity sensor that was made of $CeO_2$ nanowires. However, no attempt has been used for fabricating an impedance-type humidity sensor that was made of pure $EuCl_2$, $Eu_2O_3$ and $EuCl_2/Eu_2O_3$ blend films. In this work, the impedance-type humidity sensors that were made of the $EuCl_2$, $Eu_2O_3$ and $EuCl_2/Eu_2O_3$ blend films were fabricated. The characterization of the $EuCl_2$, $Eu_2O_3$ and $EuCl_2/Eu_2O_3$ blend films were studied using scanning electron microscopy (SEM) and X-ray diffraction (XRD). The humidity-sensing characteristics of the $EuCl_2$, $Eu_2O_3$ and $EuCl_2/Eu_2O_3$ blend films, including the response, linearity, hysteresis, response/recovery times, influence of ambient temperature, influence of applied frequency and stability, were studied. The complex impedance spectra were used to investigate the humidity-sensing mechanism of the $EuCl_2/Eu_2O_3$ blend film.

## 2. Experimental Methods

### 2.1. Materials and Humidity Sensors Preparation

Europium dichloride ($EuCl_2$, 99%, Sigma-Aldrich, St. Louis, MO, USA) was used as received without further purification. The fabrication method of europium oxide ($Eu_2O_3$) was the thermal decomposition technique that was described in the literature [28]. The starting material was $EuCl_2$ and the decomposition temperature was 600 °C for 5 h under the ambient atmosphere in furnace. The x wt% $EuCl_2$ with 2, 5, 6, 7 and 8%wt/$Eu_2O_3$ blends were prepared using a wet-blending process. The $Eu_2O_3$ particles were impregnated with aqueous solutions of various x wt% $EuCl_2$ solutions under ultrasonicating for 1 h to achieve a homogeneous dispersion of the $Eu_2O_3$ particles. Figure 1a shows the structure of an impedance-type humidity sensor. The interdigitated Au electrodes were made on an alumina substrate using a screen-printing method. The gap size and line width of the Au electrode were 0.25 and 0.2 mm, respectively. Then, 20 μL of the as-prepared uniformly $EuCl_2$, $Eu_2O_3$ and $EuCl_2/Eu_2O_3$ blend precursor solutions were drop-coated on an as-prepared alumina substrate using a micropipette, followed by drying at 110 °C.

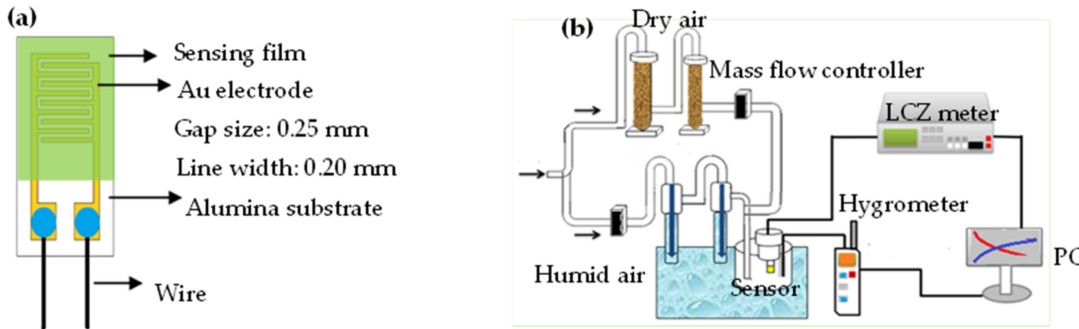

**Figure 1.** (**a**) structure of humidity sensor and (**b**) the impedance measurement of humidity sensors and humidity atmosphere controller.

### 2.2. Characterization of EuCl₂, Eu₂O₃ and EuCl₂/Eu₂O₃ Bend Films

The composition and morphologies of the $EuCl_2$, $Eu_2O_3$ and $EuCl_2/Eu_2O_3$ blend film was investigated using an X-ray diffraction (XRD) using Cu $K_\alpha$ radiation (Shimadzu, Lab XRD-6000, Taipei, Taiwan) and a scanning electron microscope (SEM, Hitachi, TM400 Plus, Tokyo, Japan).

### 2.3. Measurement of Electrical and Humidity-Sensing Properties

Figure 1b shows the electrical and humidity-sensing measurement system. The generation of required humidity conditions for testing sensors was controlled using a divided humidity generator system in a temperature-controlled testing chamber. The principal

apparatus for controlling the generation of humidity was a divided humidity generator, in which the proportion of dry and humid air under a total flow rate was 10 L/min to obtain the required humidity conditions for testing. The carrier gas was dry air. The relative humidity (RH) values were determined using the displayed readings of a standard humidity hygrometer (with an accuracy of $\pm 0.1\%$ RH). The electronic properties (impedance) of the as-prepared humidity sensors vs. RH were measured using an LCZ meter.

## 3. Results and Discussion

### 3.1. Characteristics of EuCl$_2$, Eu$_2$O$_3$ and EuCl$_2$/Eu$_2$O$_3$ Blend Films

XRD Characterization and Morphology Observations

Figure 2a shows the XRD of EuCl$_2$, the peaks appearing at $2\theta = 23.1°$, $26.1°$, $29.4°$, $32.1°$, $35.6°$, $38.1°$, $39.6°$, $41.2°$, $47.8°$, $50.9°$, $60.9°$ and $64.9°$ corresponded to the (210), (111), (211), (121), (301), (002), (230), (131), (212), (331), (232) and (610) planes of the orthorhombic structure of EuCl$_2$ [29]. Figure 2b shows the XRD spectrum of the Eu$_2$O$_3$ film that was made of the thermal decomposition of the EuCl$_2$. The peaks appearing at $2\theta = 28.5°$, $38.0°$, $42.4°$, $47.3°$, $56.0°$ and $77.0°$ corresponded to the (222), (332), (431), (440), (622) and (662) planes of the body-centered cubic (BCC) structure of Eu$_2$O$_3$, indicating the formation of Eu$_2$O$_3$ crystals [30,31]. Additionally, a very similar XRD spectrum has been reported for Eu$_2$O$_3$ prepared by using a solution method [31]. Figure 2c shows the XRD of the EuCl$_2$/Eu$_2$O$_3$ blend; the peaks show it had mixed phases of EuCl$_2$ and Eu$_2$O$_3$ and no noticeable peak shifts were observed.

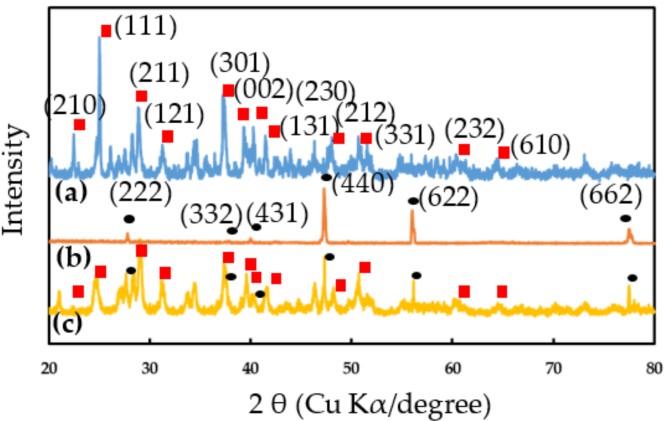

**Figure 2.** XRD patterns of (**a**) EuCl$_2$, (**b**) Eu$_2$O$_3$ and (**c**) EuCl$_2$/Eu$_2$O$_3$ blend films.

Figure 3 shows the morphology of the EuCl$_2$, Eu$_2$O$_3$ and EuCl$_2$/Eu$_2$O$_3$ blend films that were analyzed using scanning electron microscopy. Figure 3a shows the EuCl$_2$ film that had various unfixed shapes of a massive lamination structure. Figure 3b shows the Eu$_2$O$_3$ film, the Eu$_2$O$_3$ particles obviously aggregated to form a tight surface morphology. Figure 3c shows the 7 wt% EuCl$_2$/Eu$_2$O$_3$ blend film in a low-magnification image; this film had smoother surfaces than the Eu$_2$O$_3$ film did and many cracks in its surface. Figure 3d shows a high-magnification image of Figure 3c; the film exhibited porous structures marked by white arrows.

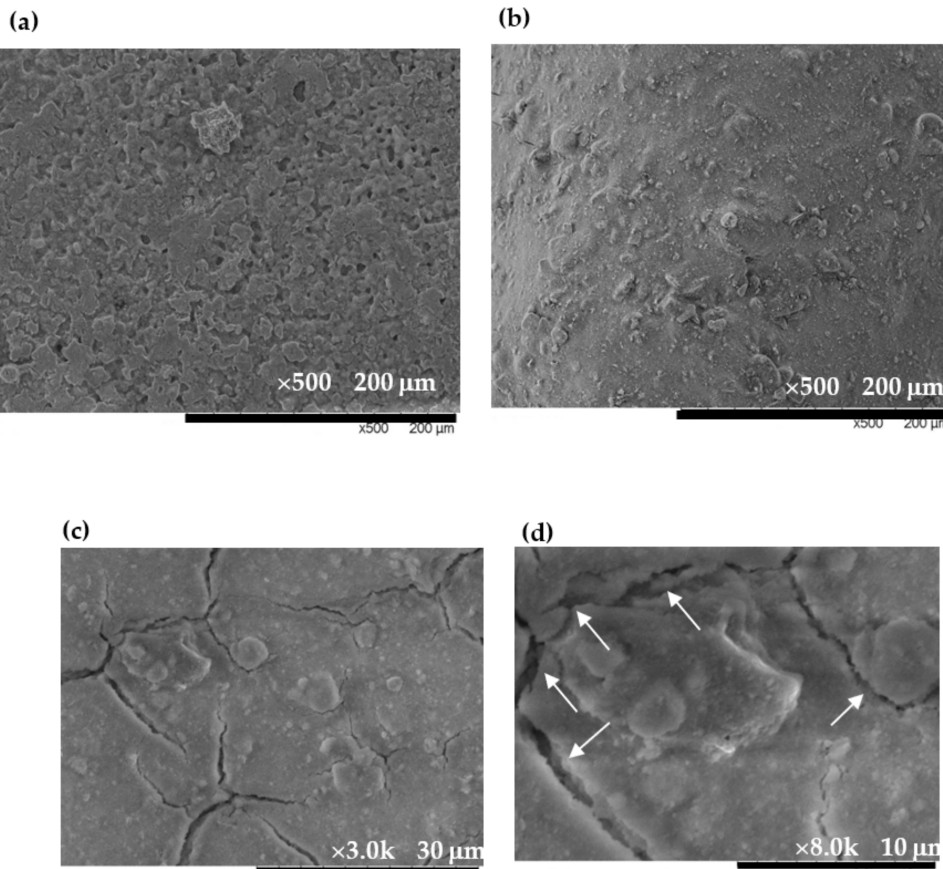

**Figure 3.** SEM micrographs of (**a**) EuCl$_2$ film, (**b**) Eu$_2$O$_3$ film, (**c**) 7 wt% EuCl$_2$/Eu$_2$O$_3$ blend film and (**d**) high-magnification image of EuCl$_2$/Eu$_2$O$_3$ blend film.

### 3.2. Electrical and Humidity-Sensing Properties of Humidity Sensors Based on EuCl$_2$, Eu$_2$O$_3$ and EuCl$_2$/Eu$_2$O$_3$ Blend Films

Figure 4 plots the log-impedance of the EuCl$_2$, Eu$_2$O$_3$ and EuCl$_2$/Eu$_2$O$_3$ blend films as a function of the relative humidity. Table 1 presents the results of the sensitivity and linearity of humidity sensing. The sensitivity and linearity were calculated as the slope and R-squared value (R$^2$) of the linear fitting curve in the humidity range from 20 to 90% RH, respectively. The EuCl$_2$ film exhibited a steep decrease in impedance as the RH changed from 20 to 40% RH, and very slowly decreased in the range of 40–90% RH. This result was related to the fact that EuCl$_2$ is very moisture sensitive [28]. The Eu$_2$O$_3$ film had one less order changed in impedance, with the humidity ranging from 40 to 90% RH and almost no impedance changed in the range of 20–40% RH because of its weak water adsorption and low-conduction properties. For obtaining the higher sensitivity and better linearity of the Eu$_2$O$_3$ film in a wider humidity range, a EuCl$_2$/Eu$_2$O$_3$ blend film was fabricated, and the optimum ratio of EuCl$_2$ to Eu$_2$O$_3$ was studied. The impedance of all the EuCl$_2$/Eu$_2$O$_3$ blend films continuously decreased along with the humidity increase in the range of 20–40% RH, suggesting that the strong water adsorption capacity of EuCl$_2$ improved the sensitivity of the EuCl$_2$/Eu$_2$O$_3$ blend film. The sensitivity (slope) of the 7 wt% EuCl$_2$/Eu$_2$O$_3$ blend film was greater than those of the 2, 5, 6 and 8 wt% EuCl$_2$/Eu$_2$O$_3$ blend films in the studied range (20 to 90% RH). This result was related to the fact that the blended amounts of EuCl$_2$ increased, which increased the water adsorption capacity for physisorption and chemisorption layers on the EuCl$_2$/Eu$_2$O$_3$ blend film with the humidity ranging from 40 to 90% RH. Additionally, the 7 wt% EuCl$_2$/Eu$_2$O$_3$ blend film had better linearity than that of the 8 wt% EuCl$_2$/Eu$_2$O$_3$ blend film because the impedance of the 8 wt% EuCl$_2$/Eu$_2$O$_3$ blend film slightly changed in the range of 40–90% RH. The 7 wt% EuCl$_2$/Eu$_2$O$_3$ blend

film exhibited the highest response and best linearity; therefore, it was further tested to investigate its humidity-sensing properties and mechanism.

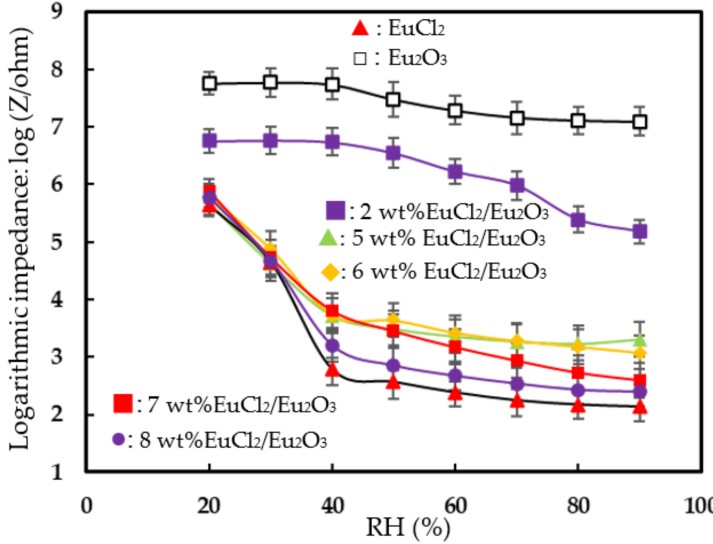

**Figure 4.** Log-impedance vs. relative humidity for humidity sensors based on $EuCl_2$, $Eu_2O_3$ and $EuCl_2/Eu_2O_3$ blend films. Measurements were made at 25 °C, 1 V AC voltage and 1 kHz frequency.

**Table 1.** Sensitivity and linearity of impedance-type humidity sensors based on $EuCl_2$, $Eu_2O_3$ and $EuCl_2/Eu_2O_3$ blend films.

| Materials | Linear Fitting Curve | Sensitivity [a] (log Z/% RH) | Linearity [b] ($R^2$) |
|---|---|---|---|
| $EuCl_2$ | Y = −0461 X + 5.612 | −0.0461 | 0.7267 |
| $Eu_2O_3$ | Y = −0118 X + 8.072 | −0.0118 | 0.9230 |
| 2 wt% $EuCl_2/Eu_2O_3$ | Y = −0245 X + 7.546 | −0.0245 | 0.8983 |
| 5 wt% $EuCl_2/Eu_2O_3$ | Y = −0296 X + 5.464 | −0.0296 | 0.6983 |
| 6 wt% $EuCl_2/Eu_2O_3$ | Y = −0346 X + 5.782 | −0.0346 | 0.7797 |
| 7 wt% $EuCl_2/Eu_2O_3$ | Y = −0427 X + 6.015 | −0.0427 | 0.8601 |
| 8 wt% $EuCl_2/Eu_2O_3$ | Y = −0411 X + 5.741 | −0.0411 | 0.7582 |

[a] Sensitivity is defined as the slope of the linear fitting curve from 20 to 90% RH. [b] Linearity is defined as the R-squared value (correlation coefficient) of the linear fitting curve from 20 to 90% RH.

Figure 5a shows the hysteresis of the 7 wt% $EuCl_2/Eu_2O_3$ blend film. The average hysteresis was below 1.1% RH as the humidity ranged from 20 to 90% RH in a desiccation-to-humidification cycle. The reversibility was investigated with the hysteresis of testing a desiccation-to-humidification cycle at 60% RH three times. The reversibility was 1.07% RH. Figure 5b shows the influence of ambient temperature on the impedance of the 7 wt% $EuCl_2/Eu_2O_3$ blend film vs. RH. The average temperature coefficient was about −0.10% RH/°C. Figure 5c shows the response/recovery times of the 7 wt% $EuCl_2/Eu_2O_3$ blend film. The response/recovery times were 40/80 s. The response/recovery times of the $EuCl_2$ film were 30/140 s. The 7 wt% $EuCl_2/Eu_2O_3$ blend film had faster response/recovery times than that of the $EuCl_2$ film. This result was related to the strong water adsorption capacity of $EuCl_2$. Figure 5d shows the influence of the applied frequency on the impedance of the 7 wt% $EuCl_2/Eu_2O_3$ blend film vs. RH. The applied frequency affected the impedance at low humidity more significantly (<40% RH) than that at high humidity. Figure 5e plots the long-term stability of the 7 wt% $EuCl_2/Eu_2O_3$ blend film. At testing points of 20, 60 and 90% RH, no obvious deviations in impedance were found within 53 days. The repeatability, on the same day, was performed by repeating testing at 60% RH three times and analyzed with relative standard deviation (RSD). The repeatability (RSD) was 6.3%. The humidity-sensing properties of this study were compared with those humidity sensors

that were made of ceramic materials in the literature [31–33], as shown in Table 2. The present humidity sensor that was made of the 7 wt% $EuCl_2/Eu_2O_3$ blend film using a simple thermal decomposition technique had a wide humidity-sensing range, a comparable sensitivity and low hysteresis compared to the humidity sensors that were made of $Li^+$ and $K^+$ ions-doped ZnO, $SnO_2$ and $TiO_2$.

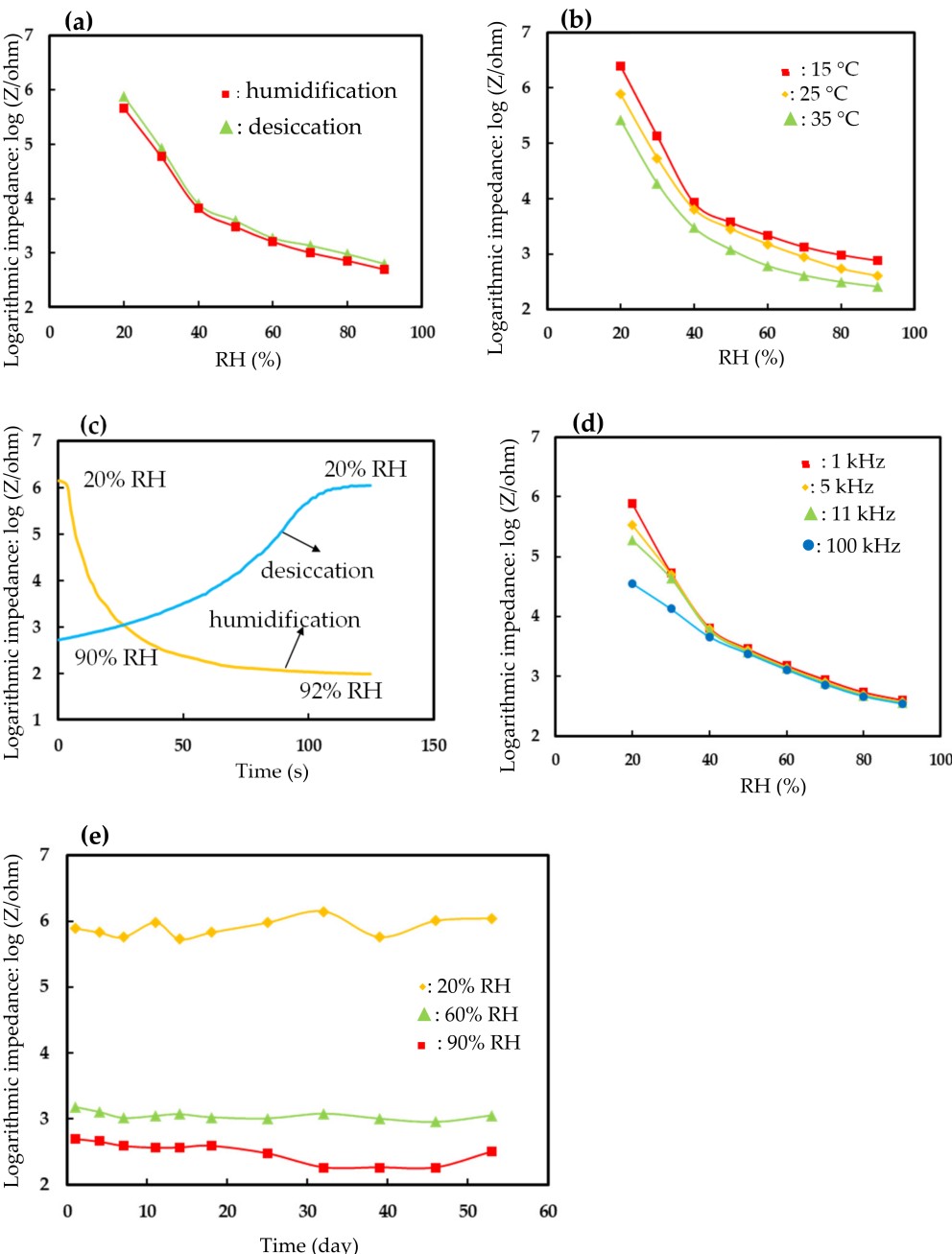

**Figure 5.** Humidity-sensing properties of the humidity sensor based on 7 wt% $EuCl_2/Eu_2O_3$ blend film. (**a**) Hysteresis, (**b**) effect of ambient temperature, (**c**) response/recovery times, (**d**) effect of applied frequency, (**e**) long-term stability.

**Table 2.** Humidity sensor performance of this work compared with the humidity sensors based on ceramic materials in the literatures.

| Sensing Material | Working Range (% RH) | Sensitivity | Hysteresis (% RH) | Response Time (s) | Ref. |
|---|---|---|---|---|---|
| LiCl-doped $TiO_2$ | 13–65 | — | — | 0.5 | [32] |
| LiCl-doped ZnO | 11–95 | — | 2 | 3 | [33] |
| KCl-doped $SnO_2$ | 11–95 | 4 order [a] | — | 5 | [34] |
| $EuCl_2$-blended $Eu_2O_3$ | 20–90 | 0.0427 [b] | <1.1 | 40 | This work |

[a] Sensitivity is defined as order in impedance cganges over entire testing humidity range. [b] Sensitivity is defined as slope ($-(\log Z/\%$ RH)) of the linesr fitting curve over entire testing humidity range.

### 3.3. Humidity-Sensing Mechanism

The complex impedance spectrum was useful for studying the conduction mechanisms of humidity sensors. Figure 6 shows the measured impedance spectra of the humidity sensor that was made of the 7 wt% $EuCl_2/Eu_2O_3$ blend film. At low humidity (20% RH), a semicircular plot of the film impedance was obtained. The semicircle plot of the impedance has been explained by many authors [35–37], resulting mainly from the intrinsic impedance of the 7 wt% $EuCl_2/Eu_2O_3$ blend film, and the film could be modeled as an equivalent parallel circuit that incorporates a resistor and a capacitor. When increasing the RH (40, 50 and 60%), the semicircle radius gradually reduced and a straight line appeared at low frequencies. The straight line represented Warburg impedance, which was caused by the diffusion of $H_3O^+$ ions across the interface between the electrode and the sensing film [35]. Finally, when increasing the RH to 80%, the semicircle disappeared and only a straight line was observed. These results were related to the fact that, upon the adsorption of water, the adsorbed water molecules on the 7 wt% $EuCl_2/Eu_2O_3$ blend film formed a thin liquid layer, and resultingly, gradually dissociated to form $H_3O^+$ ions. At high RH, the sorbed water acted as a plasticizer, increasing the mobility of the solvated $H_3O^+$ ions diffusing across the interface between the electrode and the liquid-like 7 wt% $EuCl_2/Eu_2O_3$ blend film. According to the obtained complex impedance plots, the humidity-sensing by the 7 wt% $EuCl_2/Eu_2O_3$ blend film depended on the $H_3O^+$ ion transport mechanism [37,38].

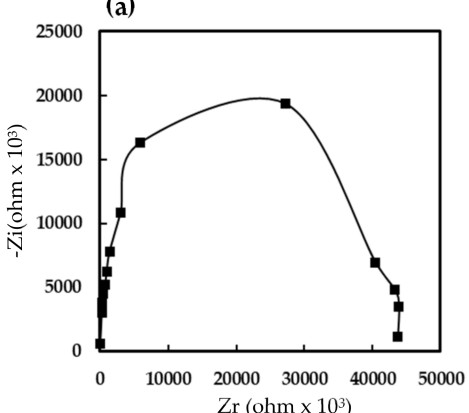 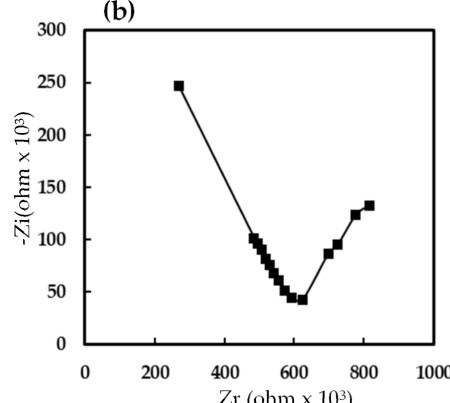

**Figure 6.** *Cont.*

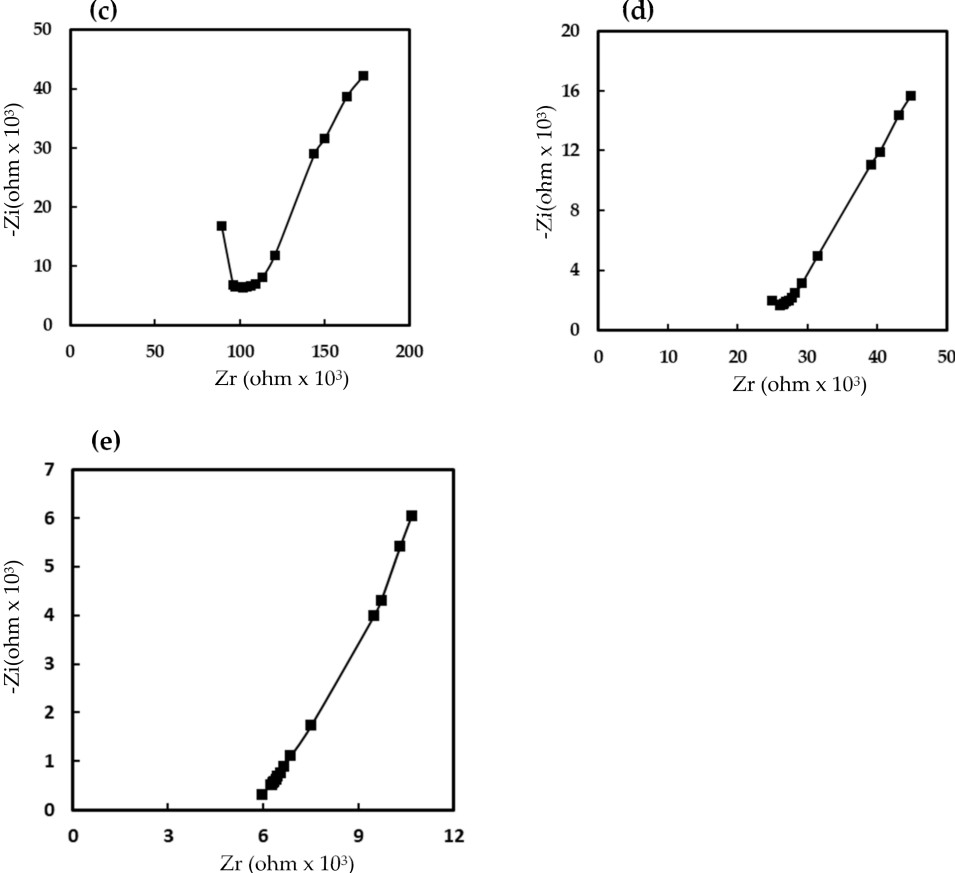

**Figure 6.** Complex impedance plots of humidity sensor based on 7 wt% $EuCl_2/Eu_2O_3$ blend film at (**a**) 20% RH, (**b**) 40% RH, (**c**) 50% RH, (**d**) 60% RH and (**e**) 80% RH. Measurements were made at frequency ranging from 50 to 100,000 Hz, RH ranging from 20 to 80% RH, at 1 V AC voltage and at 25 °C.

## 4. Conclusions

The humidity sensor based on the $EuCl_2$ film was well suited to low humidity (20–40% RH) because of its strong water adsorption property. The humidity sensor based on the $Eu_2O_3$ film exhibited a small humidity-working range (40~90% RH) because of its weak water adsorption and low-conduction properties. The humidity sensor based on the $EuCl_2/Eu_2O_3$ blend film exhibited high sensitivity and good linearity over the entire RH range (20 to 90% RH) because of the added $EuCl_2$ to increase the water adsorption and conductance of the $EuCl_2/Eu_2O_3$ blend film. The impedance-type humidity sensor that was made of the 7 wt% $EuCl_2/Eu_2O_3$ blend film exhibited high sensitivity (slope = 0.0427) and the best linearity ($R^2$ = 0.8601), small hysteresis (<1.1% RH), a small ambient temperature coefficient (−0.10% RH/ °C), fast response/recovery times (40/80 s) and good long-term stability (at least 53 days). The complex impedance plots of the $EuCl_2/Eu_2O_3$ blend film changed from semicircular to linear as the RH increased. These results reflect the $H_3O^+$ ions that dominated the conductance of the $EuCl_2/Eu_2O_3$ blend film.

**Author Contributions:** Conceptualization, P.-G.S.; methodology, P.-G.S.; investigation, P.-G.S. and N.-H.C.; writing—original draft preparation, P.-G.S.; writing—review and editing, P.-G.S.; supervision, P.-G.S.; project administration, P.-G.S.; funding acquisition, P.-G.S. All authors have read and agreed to the published version of the manuscript.

**Funding:** This research was founded by Ministry of Science and Technology of Taiwan, grant no. MOST 110-2113-M-034-002.

**Institutional Review Board Statement:** Not applicable.

**Informed Consent Statement:** Not applicable.

**Data Availability Statement:** Not applicable.

**Conflicts of Interest:** The authors declare no conflict of interest.

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
