# Peer review of "Electrical and Humidity-Sensing Properties of EuCl2, Eu2O3 and EuCl2/Eu2O3 Blend Films"

_chemosensors, doi:10.3390/chemosensors9100288_

Round 1

Reviewer 1 Report

The authors compared the performance of humidity sensors based on EuCl2, Eu2O3 and EuCl2/Eu2O3 through experiments, and found that EuCl2/Eu2O has a better humidity-sensing property. The result is interesting. The paper may be accepted after solving the following comments:

  1. The English writing has to be improved. There are some irregularities and errors in English writing. For example, both “Fig.” and “Figure” are used in the article.
  2. I recommend trying to avoid word of “novel”.
  3. All the pictures are very vague, it is recommended that the author use clearer pictures.
  4. The color of the picture is single and difficult to distinguish. It is recommended that the author modify the picture according to the literature (ACS Appl. Mater. Interfaces 2021, 13, 23).
  5. How about the reversibility and repeatability of the sensors?
  6. In addition to impedance type humidity sensors, there are many other types of humidity sensors, such as resistive humidity sensors, capacitive sensors, optical fiber humidity sensors, and QCM humidity sensors. It is suggested to add relevant discussion and references more references for background information and performance comparison. (ACS Appl. Mater. Interfaces 2021, 13, 23, Sens. Actuators B Chem., vol. 341, pp. 129992, Aug. 2021. Sens. Actuators B Chem., vol. 328, pp. 129049, IEEE Sensors Journal, doi: 10.1109/JSEN.2021.3109446.)
  7. The author compared the performance of 5%wt, 6%wt, and 7%wt of EuCl2/Eu2O3, and found that 7%wt of EuCl2/Eu2O3 has better humidity sensitivity performance. Does this mean that the higher the concentration of EuCl2, the higher the sensitivity? How about the performance of 8%wt?

Author Response

1. The English writing has to be improved. There are some irregularities and errors in English writing. For example, both “Fig.” and “Figure” are used in the article.

Author reply:

We thank the reviewer’s suggestion. The “Fig.” was changed to “Figure”. Grammatical and writing style errors in the original version have been corrected by our colleague who is a native English speaker.

2. I recommend trying to avoid word of “novel”.

Author reply:

We thank the reviewer’s suggestion. The “novel” in the abstract section was deleted.

3. All the pictures are very vague, it is recommended that the author use clearer pictures.

Author reply:

We thank the reviewer’s suggestion. The pictures were replotted using clear color.

4. The color of the picture is single and difficult to distinguish. It is recommended that the author modify the picture according to the literature (ACS Appl. Mater. Interfaces2021, 13, 23).

Author reply:

We thank the reviewer’s suggestion. The pictures were replotted using clear color.

5. How about the reversibility and repeatability of the sensors?

Author reply:

We thank the reviewer’s suggestion. The reversibility and repeatability of the sensor were studied and the explanation was added in p. 5, lines 229 and 230 and p. 6, lines from 240 to 242.

6. In addition to impedance type humidity sensors, there are many other types of humidity sensors, such as resistive humidity sensors, capacitive sensors, optical fiber humidity sensors, and QCM humidity sensors. It is suggested to add relevant discussion and references more references for background information and performance comparison. (ACS Appl. Mater. Interfaces2021, 13, 23, Sens. Actuators B Chem., vol. 341, pp. 129992, Aug. 2021. Sens. Actuators B Chem., vol. 328, pp. 129049, IEEE Sensors Journal, doi: 10.1109/JSEN.2021.3109446.)

Author reply:

We thank the reviewer’s suggestion. The references were added as the new references 12 to 14.

7. The author compared the performance of 5%wt, 6%wt, and 7%wt of EuCl2/Eu2O3, and found that 7%wt of EuCl2/Eu2O3has better humidity sensitivity performance. Does this mean that the higher the concentration of EuCl2, the higher the sensitivity? How about the performance of 8%wt?

Author reply:

We thank the reviewer’s suggestion. The performance of the sensor based on the 8 wt% EuCl2/Eu2O3 blend film was studied and the explanation was added in p. 4, lines from 177 to 184.

Reviewer 2 Report

The manuscript is devoted to the investigation of EuCl2, Eu2O3 and EuCl2/Eu2O3 blend films as active layers of impedance-type humidity sensors. The conducted research corresponds to the journal subject, however the following issues should be addressed.

1) Please explain the choice of EuCl2 concentration of 5, 6 and 7 %. What will happen if the concentration will be less or more.

2) How did you control the amount of the deposited compound on the substrate surface? Please, describe details of the deposition procedure.

3) What did you use as a carrier gas? Please describe in the text of the Section 2.3. The regime of supplying moist air and purging should be also described in more detail.

4) Fig. 2. The XRD pattern is given only for Eu2O3 film. The XRD images for EuCl2 and EuCl2/Eu2O3 should also be given for comparison or corresponding explanation should be given.

5) Fig. 3. What is the ration of the components in EuCl2/Eu2O3 blend in Fig. 3. How will the SEM image change when the ratio of components in the film changes?

6) The error bars on the graphs in Fig. 4 should be given? This is necessary in order to show how significant the differences in responses of the films containing different amounts of EuCl2.

7) The authors calculate the sensitivity and linearity as the slope and R-squared value (R2) of the linear fitting curve in the humidity range from 20 to 90% RH, respectively. The linear fitting curves should be presented on the corresponding graphs.

8) The advantage of EuCl2/Eu2O3 films in comparison with EuCl2 films should be described more clearly. The difference is their sensitivity to humidity is not very big. The response and recovery time of EuCl2 films should also be given for comparison.

9) The description of the mechanism is very vague. The authors mentioned that “This result can be modeled by using an equivalent circuit of a resistor and capacitor in parallel”. Please explain what you mean in relation to the EuCl2/Eu2O3 blend. Why does the change of the concentration of EuCl2 from 5 to 7% lead to change of the mechanism? This was also not clearly described in Section 3.3.

10) Please correct typos and unsuccessful expressions in the text.

Author Response

1. Please explain the choice of EuCl2concentration of 5, 6 and 7 %. What will happen if the concentration will be less or more.

Author reply:

We thank the reviewer’s suggestion. The performance of the sensors based on the 2 and 8 wt% EuCl2/Eu2O3 blend films were studied and the explanation was added in p. 4, lines from 177 to 184.

2. How did you control the amount of the deposited compound on the substrate surface? Please, describe details of the deposition procedure.

Author reply:

We thank the reviewer’s suggestion. 20 mL of the as-prepared uniformly precursor solutions were drop-coated on an as-prepared alumina substrate using micropipet. The explanation was added in p. 2, lines from 68 to 70.

3. What did you use as a carrier gas? Please describe in the text of the Section 2.3. The regime of supplying moist air and purging should be also described in more detail.

Author reply:

We thank the reviewer’s suggestion. The explanation about the carrier gas and the supplying air was added in p. 2, lines from 78 to 82.

4. Fig. 2. The XRD pattern is given only for Eu2O3film. The XRD images for EuCl2 and EuCl2/Eu2O3 should also be given for comparison or corresponding explanation should be given.

Author reply:

We thank the reviewer’s suggestion. The XRD of the EuCl2 and EuCl2/Eu2O3 films were studied and the explanation was added in p. 3, lines from 100 to 102 and lines 107 and 108.

5. Fig. 3. What is the ration of the components in EuCl2/Eu2O3blend in Fig. 3. How will the SEM image change when the ratio of components in the film changes?

Author reply:

We thank the reviewer’s suggestion. The explanation about the ration of the components in EuCl2/Eu2O3 blend in Fig. 3 was added in caption of Fig. 3. The added amount of EuCl2 increased, the EuCl2/Eu2O3 blend film was tight.

6. The error bars on the graphs in Fig. 4 should be given? This is necessary in order to show how significant the differences in responses of the films containing different amounts of EuCl2.

Author reply:

We thank the reviewer’s suggestion. The error bars on the graphs in Fig. 4 were added.

7. The authors calculate the sensitivity and linearity as the slope and R-squared value (R2) of the linear fitting curve in the humidity range from 20 to 90% RH, respectively. The linear fitting curves should be presented on the corresponding graphs.

Author reply:

We thank the reviewer’s suggestion. The linear fitting curves were added in Table 1.

8. The advantage of EuCl2/Eu2O3films in comparison with EuClfilms should be described more clearly. The difference is their sensitivity to humidity is not very big. The response and recovery time of EuCl2 films should also be given for comparison.

Author reply:

We thank the reviewer’s suggestion. The explanation about the response and recovery time of EuCl2 film was added in p. 5, lines from 233 to 235.

9. The description of the mechanism is very vague. The authors mentioned that “This result can be modeled by using an equivalent circuit of a resistor and capacitor in parallel”. Please explain what you mean in relation to the EuCl2/Eu2O3 Why does the change of the concentration of EuCl2from 5 to 7% lead to change of the mechanism? This was also not clearly described in Section 3.3.

Author reply:

We thank the reviewer’s suggestion. The clear explanation about the humidity-sensing mechanism was added in p. 7, lines from 320 to 329 and p. 8, lines from 330 to 333.

10. Please correct typos and unsuccessful expressions in the text.

Author reply:

We thank the reviewer’s suggestion. The Grammatical and writing style errors in the original version have been corrected by our colleague who is a native English speaker.

Round 2

Reviewer 1 Report

The authors have make a significant changes in the manuscript to addressed the reviewer's questions and comments. Consequently, the final manuscript merits publication.

Reviewer 2 Report

The authors have answered all the Reviewer's comments. The manuscript can be accepted.